# LMGenDrive: LLM Reasoning Meets World Models for End-to-End Driving

## Abstract

Recent years have witnessed remarkable progress in autonomous driving, yet generalization to long-tail and open-world scenarios remains the primary bottleneck for large-scale deployment. To address this, one line of research explores LLMs and VLMs for their vision-language understanding and reasoning capabilities, equipping AVs with the ability not only to interpret rare and safety-critical situations when generating driving actions. In parallel, another line investigates generative world models to capture the spatio-temporal evolution of driving scenes, enabling agents to imagine and evaluate possible futures before acting. Inspired by human intelligence, which seamlessly unites understanding and imagination as a hallmark of AGI, this work explores a unified model that brings these two capabilities together for autonomous driving. We present LMGenDrive, the first framework that unifies LLM-based multimodal reasoning with generative world models for end-to-end closed-loop autonomous driving. Given multi-view camera inputs and natural-language instructions, our model generates both realistic future driving videos and corresponding control signals. By coupling an LLM with generative video capabilities, LMGenDrive gains complementary benefits: future video prediction enhances the LLM's spatio-temporal scene understanding, while the LLM itself provides reasoning and instruction-following capabilities. A progressive three-stage training strategy—ranging from vision pretraining to multi-step long-horizon driving—is proposed to further improve stability and performance. The resulting model can also operate in two complementary modes: low-latency online planning and autoregressive offline video generation. Experiments show that LMGenDrive significantly outperforms state-of-the-art methods on challenging closed-loop driving benchmarks, improving instruction following, spatio-temporal reasoning, and robustness to rare scenarios. Our work not only sets a new state-of-the-art in autonomous driving, but also demonstrates that unifying multimodal understanding and generation offers a foundational new paradigm toward achieving embodied AGI.

## 1 Introduction

Remarkable progress in autonomous driving has been witnessed in recent years with an increasing number of commercial autonomous vehicles (AVs) deployed on public roads. Amidst this momentum, end-to-end autonomous driving has emerged as a particularly vibrant research direction. Unlike traditional modular pipelines that separately handle perception, prediction, and planning with handcrafted interfaces, end-to-end models provide a holistic paradigm with potential to remove information bottlenecks among modules, better align model optimization with system-level performance, and scale effectively with large amounts of driving data.

Despite this progress, the problem of generalization remains the central bottleneck for the entire autonomous driving community. As we approach the frontier of real-world deployment, the ability to robustly handle long-tail edge cases and operate in open-world settings remains the defining challenge for AV systems. These scenarios can include rare but safety-critical events, distribution shifts across regions, adversarial weather conditions, as well as complex social interactions and ambiguous intent among agents. This challenge manifests across the autonomy stack: perception systems struggle to identify open-set entities, while prediction and planning models falter in extrapolating to

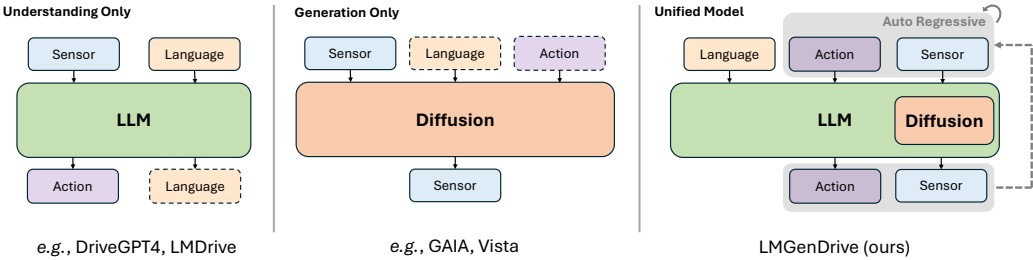

Figure 1: Comparison between existing works and ours. Prior works either leverage LLMs/VLMs for multimodal understanding and reasoning, or adopt world models for video-based scene imagination, but treat these capabilities in isolation. In contrast, our proposed **LMGenDrive** unifies both within a closed-loop end-to-end framework: the LLM interprets and reasons over multimodal inputs, while the world model simulates future scene evolution, together enabling instruction-guided planning, spatio-temporal reasoning, and robust long-horizon driving.

nondeterministic and previously unseen behaviors. These generalization failures represent the last barrier between research prototypes and truly scalable, globally deployable autonomous vehicles.

Amid this backdrop, large language models (LLMs) have emerged to demonstrate unprecedented reasoning and generalization abilities that approach—if not exceed—human-level performance. Recent models such as GPT-5 and DeepSeek-R1 (Guo et al., 2025) showcase robust capabilities in commonsense reasoning, abstraction, and decision-making. Meanwhile, vision-language models (VLMs) further extend this capacity to the multimodal domain, enabling unified interpretation of textual and visual inputs (Wang et al., 2025b; Bai et al., 2025; Liu et al., 2023; Li et al., 2022; Alayrac et al., 2022). Inspired by these vision-language understanding and reasoning capabilities, a wave of research has begun exploring how to equip AV systems with LLMs and VLMs to address the open-world, long-tail challenges in autonomous driving. As exemplified in works such as LMDrive (Shao et al., 2024) and GPT-Driver (Mao et al., 2023a), these models act as the cognitive brains to interpret ambiguous scenarios and guiding complex decision-making under uncertainty. However, most existing LLM- or VLM-empowered driving methods follow the paradigm that maps inputs directly to actions, falling short in explaining and capturing the temporal evolution of driving scenes—an essential factor for robust and anticipatory planning.

Meanwhile, another stream of research, world model (Ha & Schmidhuber, 2018), has emerged to simulate the spatio-temporal evolution of the scenes, as exemplified by video-based works such as Genie-3 (Ball et al., 2025) and Pandora (Xiang et al., 2024). Their potential has also been actively explored in the autonomous driving domain, enabling the agent to "imagine" different futures before committing to a plan. However, existing works either focus on solely generating high-fidelity scenes (Hu et al., 2023a; Russell et al., 2025; Gao et al., 2023; 2024; Ji et al., 2025; Wang et al., 2024a; Yang et al., 2024), or utilize world models as a plug-and-play forecasting module to rank multiple possible plans (Wang et al., 2024b; Wang & Peng, 2025). The integration of joint video generation and motion planning remains underexplored, limiting their ability to address critical challenges such as cumulative control errors, human-robot interaction, and the temporal consistency between generated actions and videos—factors that are essential for long-horizon problem solving in real-world systems. Moreover, these models generally lack the rich reasoning priors and instruction following capabilities uniquely offered by LLMs.

In contrast to existing models that specialize in either understanding or generation, human intelligence is inherently capable of both understanding the present and imagining the future—a dual capacity for perception and generation that underpins commonsense reasoning and long-horizon decision-making, suggesting a natural path toward artificial general intelligence (AGI). While recent studies (Deng et al., 2025; Shi et al., 2024; Chen et al., 2025; Liao et al., 2025) have shown encouraging results synergizing multimodal understanding and generation within a single model, whether this principle extends to embodied agents—and autonomous driving in particular—remains an open challenge. In this work, we propose LMGenDrive, the first framework that unifies LLM-based multimodal understanding with generative world models for closed-loop end-to-end autonomous driving. Our unified model takes multi-view camera data and natural-language driving instructions as inputs,

generating both multi-view future driving videos and control signals for the following timesteps. Concretely, we integrate an LLM and a diffusion-based video generation model: the LLM interprets and fuses visual observations with language instructions, producing learnable queries that capture the evolving scene states, which then serve as conditioning signals for the diffusion model to generate realistic multi-view driving futures. Within this unified architecture, video generation enhances the LLM's spatio-temporal scene understanding, while the LLM imparts instruction-following and reasoning capabilities to the world model—together yielding stronger and more robust closed-loop driving performance.

To enable such a unified model, we also propose a curriculum three-stage training strategy for enhanced performance and stability. First, we pretrain a vision encoder for robust driving scene understanding. Next, the frozen encoder is integrated with the LLM and video generator, and fine-tuned on single-step prediction to ground instruction following and immediate action outcomes. Finally, training is extended to multi-step sequences, enhancing long-horizon reasoning and temporal modeling for continuous driving scenarios. Once trained, the model can be applied in two modes: (1) Online planning mode: the model solely predicts planning outputs, with the diffusion generation component discarded to reduce latency; (2) Offline data generation mode: the model conducts autoregressive video generation, where the generated video and predicted control signal serve as input for the next timestep, enabling extended and consistent driving video sequences.

To sum up, our contributions are threefold: (1) *Unified closed-loop framework*. We present LM-GenDrive, the first framework that unifies LLM-based multimodal understanding with generative world models for closed-loop end-to-end autonomous driving, bridging perception, reasoning, and imagination within a single architecture; (2) *Progressive training and dual modes*. We introduce a three-stage training pipeline—from vision pretraining in driving domain, to long-horizon multi-step driving—and support two usage modes: online planning for low-latency operation, and offline autoregressive video generation for extended sequences. (3) Through comprehensive experiments, LMGenDrive achieves state-of-the-art closed-loop performance on challenging autonomous driving benchmarks, improving instruction-following, spatio-temporal reasoning, and robustness to long-tail scenarios. Beyond performance gains, it provides experimental evidence that unifying multimodal understanding and generation yields complementary benefits, pointing toward a promising path for embodied AGI.

## 2 RELATED WORKS

### 2.1 END-TO-END DRIVING

Much progress has been made in end-to-end autonomous driving, with many recent methods based on imitation learning. UniAD (Hu et al., 2023b) unified full-stack driving tasks through query-based interfaces, while ThinkTwice (Jia et al., 2023b) retrieved critical-region information to refine predictions. InterFuser (Shao et al., 2023a) used transformers to fuse multi-modal, multi-view sensor data for richer scene understanding. ReasonNet (Shao et al., 2023b) leveraged both temporal and global information of the driving scene to enhance perception, particularly in occlusion scenarios. Para-Drive (Weng et al., 2024) proposed a fully parallel architecture with a shared BEV representation, and DriveTransformer (Jia et al., 2025) went further by discarding BEV features and using pure transformers to aggregate sensor and query information. Diffusion models have also emerged for modeling diverse driving behaviors. DiffusionPlanner (Zheng et al., 2025) applied a diffusion-based policy for flexible, personalized driving, and DiffAD (Wang et al., 2025a) formulated perception and decision-making as conditional image generation. Despite these advances, most approaches still struggle with rare corner cases and lack the reasoning ability needed to generalize beyond the training distribution.

### 2.2 MLLM FOR AUTONOMOUS DRIVING

Recent advances in large language models (LLMs) (Guo et al., 2025; Yang et al., 2025a; Touvron et al., 2023a;b; Jaeger et al., 2023a; Jia et al., 2023a) and vision–language models (VLMs) (Bai et al., 2025; Zhu et al., 2023; Liu et al., 2023; Wang et al., 2025b) have motivated integrating MLLMs into autonomous driving for stronger reasoning and explainability. Early works like GPT-Driver (Mao et al., 2023a) and LanguageMPC (Sha et al., 2023) convert driving scenes into textual inputs for

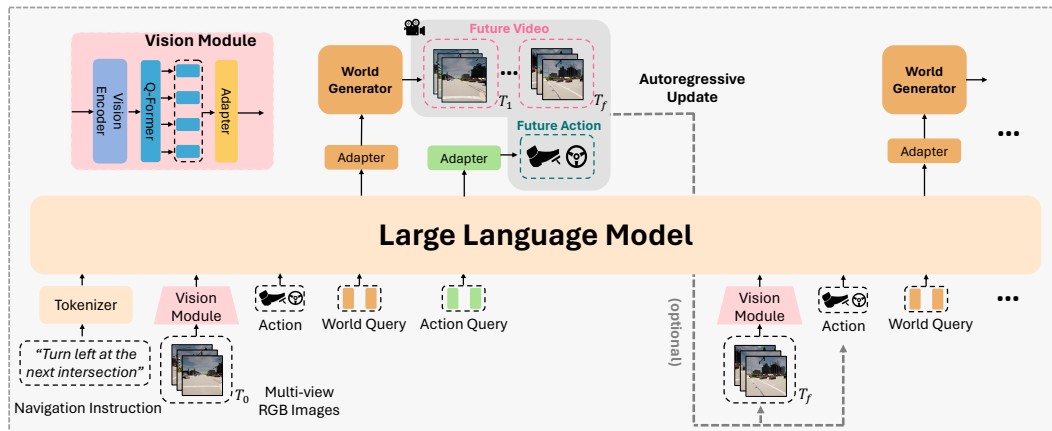

Figure 2: Overview of our unified understanding and generation architecture. We start by encoding the language instruction, multi-view RGB images, and the current action into the LLM. Two sets of learnable queries, world query and action query, are then fed into the LLM, and ultimately used to generate the future driving video and corresponding actions. The framework supports two operation modes: (1) offline data generation mode, an autoregressive generation process is adopted, where the last frame of the future video and the predicted action are used as inputs for the next timestep; (2) online planning mode, real-world data are provided as inputs for the following timestep.

direct reasoning. Later methods employ VLMs to process images and videos: some focus on visual question answering for scene understanding and optional action output (*e.g.*, DriveLM (Sima et al., 2024), DriveGPT4 (Xu et al., 2023), DriveVLM (Tian et al., 2024)), while others predict driving actions end-to-end (*e.g.*, LMDrive (Shao et al., 2024), DriveMoE (Yang et al., 2025b), BEV-Driver (Winter et al., 2025)). Agentic designs with hierarchical control, tool use, and memory, such as Agent-Driver (Mao et al., 2023b) and AD-H (Zhang et al., 2024), further extend capability. However, most MLLM-based approaches emphasize planning or explanation and lack robust modeling of how scenes and surrounding objects evolve over time—a key requirement for anticipating events and ensuring safe, long-horizon decision-making.

## 2.3 WORLD MODELS FOR AUTONOMOUS DRIVING

The concept of a *world model*, a predictive model that simulates environment dynamics, has regained attention. Video generation has become a leading paradigm, supported by advances in generative modeling, large-scale video datasets, and wide applicability. In autonomous driving, temporally grounded video prediction provides rich context for understanding and decision-making. Several methods treat pure video generation as world modeling. GAIA (Hu et al., 2023a) conditions generation on image, text, and action inputs. GAIA-2 (Russell et al., 2025) extends this to multi-view scenes, and MagicDrive (Gao et al., 2023) adds control signals such as HD maps and bounding boxes. Vista (Gao et al., 2024) scales to internet-scale driving data, while CoGen3D (Ji et al., 2025) predicts 3D-consistent representations before video synthesis to improve spatial coherence. Beyond pure generation, DriveWM (Wang et al., 2024b) predicts alternative futures for conditional planning. More recent work—DriveDreamer (Wang et al., 2024a), GenAD (Yang et al., 2024), and Prophet-DWM (Wang & Peng, 2025)—jointly models videos and actions but still mainly uses open-loop settings without direct feedback. The most related work is LAW (Li et al., 2024), which combines world modeling with closed-loop planning. However, it supervises the world model only with next-frame hidden features instead of full video generation, limiting its ability to simulate or create synthetic data. It also lacks language model integration and thus cannot handle instruction following, natural language grounding, or interactive human–AI communication in driving. To our knowledge, this is the first framework to unify LLM-based commonsense reasoning with video world models for closed-loop end-to-end autonomous driving.

## 3 METHOD

### 3.1 OVERALL FRAMEWORK

In this work, we propose LMGenDrive, a framework that unifies textual understanding/reasoning, future scene generation, and end-to-end planning. As illustrated in Figure 2, LMGenDrive is composed of three major components: (1) a vision encoder that processes multi-view camera sensor data for scene understanding and generating visual tokens; (2) a large language model and its associated component (tokenizer, Q-Former, and adapters) that takes in the language instruction, input visual tokens, world queries, and action queries, to predict the future driving scenes and actions; (3) a multi-view world generator that takes future scene tokens from the LLM and multi-view images from the last frame as inputs, to generate future multi-view driving videos. We will introduce the vision encoder in Section 3.2, the LLM with its associated components in Section 3.3, and the multiview world generator in Section 3.4. Finally, we describe the training recipe in Section 3.5.

### 3.2 VISION ENCODER

The vision encoder is designed to perceive the environment by processing, fusing, and transforming sensor data into visual tokens that can be consumed by the language model. Prior works (Shao et al., 2024; Jaeger et al., 2023b) typically leverage both multi-view images and LiDAR sensor inputs, where the LiDAR inputs are encoded into bird's-eye view (BEV) queries to extract information from multi-view images. However, our setting focuses on autoregressive video generation—where LiDAR is only available at the current frame but not in future frames. As a result, we replace LiDAR inputs with BEV positional encodings, enabling effective perception while maintaining compatibility with future video generation. The vision encoder consists of three parts: (1) In the sensor encoding part, for each image input, a 2D backbone Resnet (He et al., 2016) is applied to extract the image feature map, which is flattened to one-dimensional tokens. Tokens from different views are then fused by a transformer encoder. (2) In the BEV decoder, BEV position encodings serve as $H \times W$ queries to attend to the multi-view image features and generate BEV tokens. In addition, the learnable queries and one extra query generate corresponding waypoint tokens and one traffic-light token, respectively. The three types of visual tokens (BEV, waypoint, and traffic light) will be presented to the LLM, providing rich scene information. (3) Lastly, as the first-stage training, the vision encoder is pretrained on perception tasks (BEV object detection, traffic light recognition, waypoint prediction) by feeding the three types of tokens to additional prediction heads. Three loss terms, including the detection loss (Shao et al., 2023b), the $l_1$ waypoint loss and the cross-entropy traffic light prediction loss, are applied respectively. Note that, following LMDrive (Shao et al., 2024), once pretrained, these prediction heads are discarded and the encoder is frozen, serving as the vision encoder for the large language model.

### 3.3 LLM FOR INSTRUCTION-FOLLOWING DRIVING AND SCENE UNDERSTANDING

As depicted in Figure 2, our system casts the LLM as the "brain" of the entire driving pipeline: it ingests sensor tokens emitted by the frozen vision encoder at every frame and parses natural-language commands, to forecast upcoming maneuvers and emits conditioning features for subsequent video generation. We adopt LLaMA (Touvron et al., 2023a) as the linguistic architecture due to its broad success in both language-centric (Zheng et al., 2023; Geng et al., 2023) and vision-grounded (Liu et al., 2023; Zhu et al., 2023) instruction-tuning settings.

**Instruction and visual tokenization.** As the model takes navigation instruction and multi-view image as inputs, their tokenization is our first step. For the navigation instruction, we tokenize them with the LLaMA tokenizer (Touvron et al., 2023a). For the multi-view images, each frame is tokenized by the aforementioned vision encoder, and the resulting tokens are buffered together with the most recent token history (up to $T_{\max}$ frames) to curb cumulative error and maintain temporal coherence during executing the driving instruction in the closed loop. For each frame, the pretrained vision encoder outputs $H \times W$ BEV tokens, 4 waypoint tokens, and 1 traffic-light token. Passing all visual tokens (about 2k per frame) to the LLM is computationally prohibitive. To compress them, we use a Q-Former with 8 learnable queries per frame that attend to the raw tokens and distill them into compact frame-level features. An MLP adapter then projects these features to the LLM's embedding dimension for seamless fusion with language tokens.

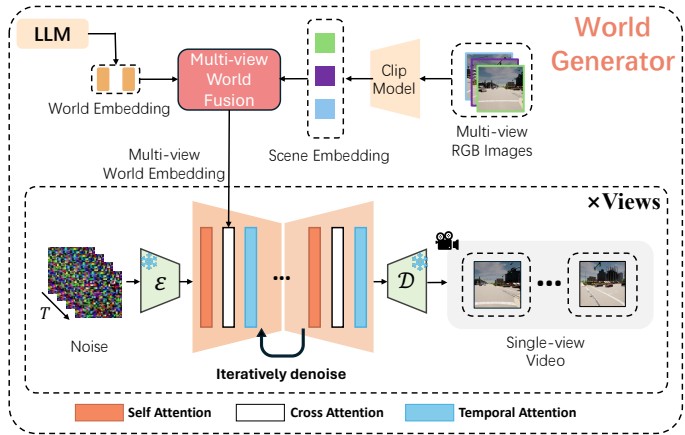

Figure 3: Architecture of our world generator. We begin by fusing the world embedding obtained from the LLM with multi-view RGB images. The fused multi-view world embedding is then injected into the diffusion model with the cross-attention mechanism to generate multi-view future videos.

**Action prediction.** Together with the instruction and visual tokens, we feed $N$ learnable action query tokens into the LLM. After passing through the LLM and associated adapters, these queries evolve into $N$ latent feature vectors, each encoding the spatio-temporal context needed for motion planning. A subsequent two-layer MLP maps these $N$ feature vectors to $N$ predicted waypoints and outputs a binary flag indicating whether the current instruction has been completed. Finally, the predicted waypoints are converted into low-level control commands—brake, throttle, and steering—through two independent PID controllers (Chen et al., 2020) that separately regulate longitudinal velocity and lateral heading, ensuring accurate trajectory tracking.

## 3.4 Multi-View World Model

While the LLM is responsible for instruction following and reasoning, autonomous driving also requires modeling the visual dynamics of the environment. To this end, we introduce a multi-view world model that generates future video frames conditioned on the LLM outputs. By aligning action predictions with video generation, our framework jointly reasons about both the agent's behavior and the evolution of the surrounding world. Our video generation process additionally *supports* an autoregressive mode (Xiang et al., 2024), which can be optionally enabled during inference: the future action and frame predicted at the previous timestep can be fed back into the LLM as input for the next prediction.

**World Query Conditioning.** As shown in Figure 2, in addition to the instructional tokens, visual tokens, and action query mentioned above, the LLM also takes the world query as input. Passing through the LLM, these world queries aggregate information from instructions, sensor inputs, and the actions, thereby enabling the model to form an internal representation of the world dynamics. Conceptually, these queries serve as a bridge to the world's temporal evolution, and act as the conditioning signal fed into the following world generator to synthesize future driving videos.

**Multi-view Image Conditioning.** As shown in Figure 3, besides using scene queries to capture world evolution, we incorporate the last-frame multi-view images to supply fine-grained appearance details and the initial world state. These images are encoded by a CLIP model into semantically rich features that emphasize visual textures and appearance. During end-to-end training, these CLIP features are fused with LLM representations through attention blocks. Self-attention aggregates and aligns multi-view information into a unified space, and cross-attention injects LLM guidance. This design not only provides an appearance prior for consistent video generation but also encourages the LLM to focus on dynamic, motion-related representations.

**World Generator.** After the multi-view world fusion step, we obtain a set of multi-view world embeddings, each corresponding to one camera view. Taking these embeddings as the final conditioning feature, our world generator employs a U-Net (Ronneberger et al., 2015) diffusion architecture

to produce future frames. For each view, the associated embedding is injected into the diffusion process through a dedicated cross-attention module, ensuring that view-specific information is effectively transferred. The model follows the standard denoising diffusion process (Ho et al., 2020): starting from pure Gaussian noise and progressively removing noise to generate a future video sequence. The output is a video tensor of shape $\mathbb{R}^{v \times t \times h \times w \times 3}$, where $v$ is the number of views, $t$ is the temporal length, and $h, w$ are the spatial resolution. Inspired by (Wang et al., 2024b; Guo et al., 2023), we further augment the U-Net blocks with spatio-temporal transformers to better capture temporal dynamics and spatial structure in driving scenes.

### 3.5 TRAINING RECIPE

We adopt a three-stage training strategy to progressively build the model's perception, reasoning, and generation capabilities. This curriculum ensures stable convergence and enables effective long-horizon temporal modeling.

**Stage 1: Vision Encoder Pretraining.** We first pretrain the vision encoder on single-frame perception tasks using 3M expert-collected frames from CARLA (Shao et al., 2023a). Perception heads are attached for object detection, traffic light classification, and waypoint regression. After convergence, only the vision encoder is retained and frozen in later stages.

**Stage 2: Single-Step Planning and Generation.** Next, we jointly fine-tune the LLM and the video generator for single-step prediction. The vision encoder is frozen to reduce memory usage. The model takes as input a single-frame multi-view image, natural language instruction, action queries, and world queries, to predict the next waypoint, the instruction completion flag, and the future driving video. This stage enables the LLM to learn grounded instruction-following and understand how the world evolves under given actions. Simultaneously, the world generator learns to synthesize multi-view driving videos conditioned on the last frame and LLM-generated features.

**Stage 3: Multi-Step Long-Horizon Training.** We progressively expand training to 2–3-step sequences to strengthen long-horizon reasoning. Specifically, previously generated videos are autoregressively fed as input for the next step's generation. To save memory, the video generator is frozen while gradients still propagate, and the LLM remains fully trainable. This design encourages the LLM to capture temporal dependencies—such as other agents' intentions, speed, and interactions—over extended observation windows. As a result, the LLM develops stronger temporal abstraction and inductive reasoning abilities for dynamic driving scenes.

**Training Objectives.** We apply three loss terms in the last two stages: (1) $l_1$ waypoint regression loss; (2) binary classification loss for instruction completion; (3) diffusion loss for video generation:

$$\mathcal{L}_{\text{DM}} = \mathbb{E}_{t,\epsilon} \left[ \|\epsilon_\theta(\mathbf{z}_t, \mathbf{c}, t) - \epsilon\|^2 \right],$$

where $\mathbf{z}_t$ is the noisy latent at timestep $t$, $\epsilon$ is the added Gaussian noise, and $\mathbf{c}$ denotes conditioning features from the multi-view image and scene queries.

## 4 EXPERIMENTS

### 4.1 EXPERIMENT SETUP

**Training Details.** During training, three synchronized RGB cameras (left, front, right) are resized to $224^2$ pixels and sampled at 10 Hz, and an 8-frame temporal window is considered. The network is tasked with predicting four future waypoints at $t + \{0.2, 0.4, 0.6, 0.8\}$ s, along with eight future video frames from $t + 0.1$ s to $t + 0.9$ s in 0.1-second increments. We optimize the model using AdamW optimizer (Loshchilov & Hutter, 2018) with an initial learning rate of $1 \times 10^{-5}$ on eight NVIDIA H800 GPUs under DeepSpeed ZeRO-2; convergence is reached in roughly two days. Due to GPU-memory constraints, the third training stage operates on one to three timesteps. The system uses Vicuna-7B (Chiang et al., 2023) as the LLM backbone, Stable Diffusion 1.5 (Rombach et al., 2022) for image generation, and AnimateDiff (Guo et al., 2023) for temporal modeling.

**Benchmark.** We implement and evaluate our approach using the open-source CARLA simulator of version 0.9.10.1 (Dosovitskiy et al., 2017) on the LangAuto benchmark (Shao et al., 2024). The LangAuto benchmark comprises test routes that traverse eight CARLA towns, span diverse weather

settings, and contain deliberately misleading linguistic cues. It consists of three tracks, LangAuto, LangAuto-Short, and LangAuto-Tiny, which varies in the route length. During evaluation, each method controls the vehicle using only natural-language commands and visual observations.

**Metric.** Following the CARLA Leaderboard (CARLA Team, 2020) and LangAuto (Shao et al., 2024), we report route completion (RC), infraction score (IS), and driving score (DS). RC measures the fraction of the planned route completed before exceeding the deviation tolerance. IS penalizes collisions and traffic-rule violations via a decaying factor. DS, the product of RC and IS, serves as the primary overall indicator of safe and efficient driving. For generated videos, we further evaluate perceptual quality using Fréchet Video Distance (FVD) and Fréchet Inception Distance (FID), which assess temporal consistency and visual realism, respectively.

## 4.2 SoTA Comparison

| Methods | LangAuto | | | LangAuto-Short | | | LangAuto-Tiny | | |
|---|---|---|---|---|---|---|---|---|---|
| | DS ↑ | RC ↑ | IS ↑ | DS ↑ | RC ↑ | IS ↑ | DS ↑ | RC ↑ | IS ↑ |
| LMDrive (Shao et al., 2024) | 10.7±3.8 | 16.2±4.9 | 0.63±0.04 | 14.2±4.4 | 20.1±4.4 | 0.72±0.04 | 20.1±4.1 | 24.7±5.1 | 0.75±0.03 |
| AD-H† (*Zhang et al., 2024*) | 44.0 | 53.2 | 0.83 | 56.1 | 68.0 | 0.78 | 77.5 | 85.1 | 0.91 |
| BEVDriver (Winter et al., 2025) | 48.9 | 59.7 | 0.82 | 66.7 | 77.8 | 0.87 | 70.2 | 81.3 | 0.87 |
| Ours | **62.2±3.3** | **74.5±4.1** | **0.85±0.04** | **77.1±4.1** | **87.9±3.5** | **0.88±0.03** | **84.1±3.6** | **92.5±4.0** | **0.92±0.04** |

Table 1: Performance comparison on the LangAuto benchmark. We report the metrics for 3 evaluation runs. AD-H† leverages an extra model OPT-350M (Zhang et al., 2022) for low-level control.

The experimental results in Table 1 demonstrate that our method significantly outperforms existing state-of-the-art approaches on the LangAuto benchmark. Specifically, LMDrive (Shao et al., 2024) achieves a driving score (DS) of 10.7 in the LangAuto track, while AD-H (Zhang et al., 2024) and BEVDriver (Winter et al., 2025) demonstrates improved DS values of 44.0 and 48.9, respectively. Our method further push the performance to a higher level, with a DS of 62.2. In terms of route completion (RC) and infraction score (IS), our approach also shows superior performance across all three tracks: LangAuto, LangAuto-Short, and LangAuto-Tiny, showing the effectiveness of our method in handling more complex driving scenarios with language instructions.

## 4.3 Ablation studies

| Module design | DS ↑ | RC ↑ | IS ↑ |
|---|---|---|---|
| baseline | **62.2±3.3** | **74.5±4.1** | **0.85±0.04** |
| w/o world generator | 53.4±2.2 | 65.8±4.2 | 0.80±0.01 |
| w/o action queries | 58.7±3.1 | 70.4±3.7 | 0.84±0.02 |
| w/o visual pre-training | 54.9±4.5 | 67.1±4.5 | 0.81±0.02 |
| w/o stage-3 training | 55.6±4.5 | 68.9±4.5 | 0.80±0.02 |

Table 2: Ablation study on the module design for planning performance.

| Module design | FID↓ | FVD↓ |
|---|---|---|
| baseline | **6.3** | **286** |
| w/o multi-view fusion | 7.8 | 371 |
| world queries: 64 → 32 | 10.1 | 318 |
| world queries: 64 → 16 | 11.6 | 424 |

Table 3: Ablation study on the module design for generation performance.

**Ablation Study on Module Design.** As shown in Table 2, we conduct four ablation experiments to quantify the contribution of each key component in our proposed LMGenDrive. (1) **w/o world generator**: Removing the world generator together with its world query sharply degrades DS to 53.4, demonstrating that the multi-view world generator is crucial for enriching the LLM's understanding of spatio-temporal dynamics and strengthening future-scene reasoning. (2) **w/o action queries**: Replacing learnable action queries with an LMDrive-style autoregressive action prediction lowers DS to 58.7, indicating that explicit action queries provide more structured supervision and lead to more reliable planning. (3) **w/o visual pre-training**: Without the first-stage driving-oriented visual pre-training, DS drops to 54.9, highlighting the importance of injecting driving-specific semantics into the vision encoder to enhance downstream scene understanding. (4) **w/o stage-3 training**: Skipping the Multi-Step Long-Horizon training stage reduces DS to 55.6, confirming that long-horizon temporal modeling is essential for robust reasoning over extended driving contexts. Overall, all ablations shows degraded DS, with missing world generator or long-horizon training causing the largest drops, confirming these modules—along with visual pre-training and action queries—are vital for accurate, safe planning in LMGenDrive.

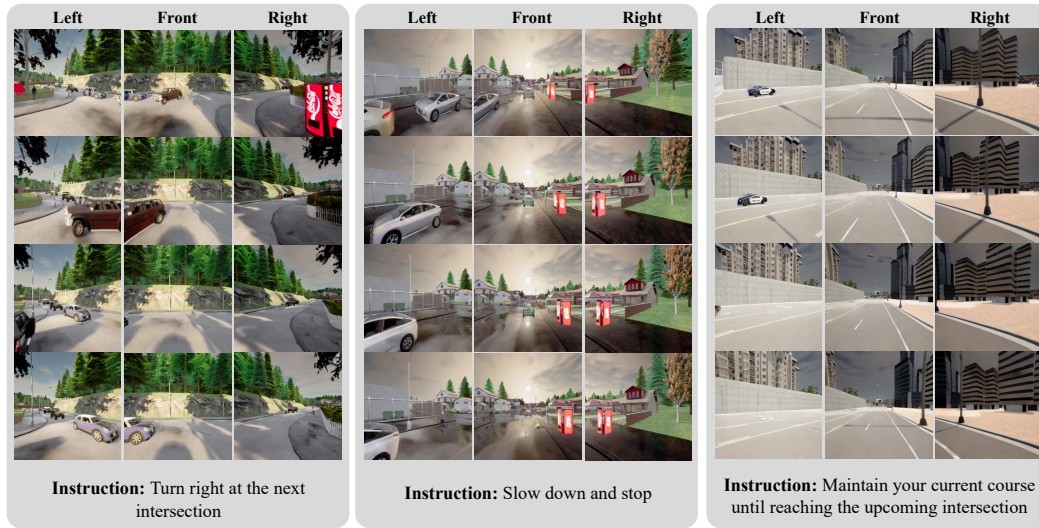

Figure 4: Visualization of multi-view future scenes generated by LMGenDrive, showing consistent left-front-right views aligned with driving instructions.

**Ablation Study on Generation Module Design.** As shown in Table 3, we further investigate how key components of the generation module influence video quality, measured by FID and FVD. (1) **w/o multi-view fusion**: Removing the cross-view fusion increases FID from 6.3 to 7.8 and FVD from 286 to 371, indicating that, without interaction among different camera views, the model struggles to maintain spatial consistency. (2) **world queries choices**: Reducing the number of world queries from 64 to 32 or 16 leads to a clear performance drop, with FID/FVD rising to 10.1/318 and 11.6/424, respectively. Overall, these results demonstrate that multi-view fusion and sufficient world queries are critical for generating coherent, high-quality videos.

## 4.4 VISUALIZATION

To illustrate LMGenDrive's capabilities, Figure 4 presents qualitative rollouts from the CARLA simulator. The top row shows the initial multi-view observations as the conditioning inputs, while the subsequent rows visualize three future steps generated by our multi-view world model in an autoregressive manner. At each step, the model takes the previously generated multi-view frames and predicts actions as input, to synthesize the next set of left, front, and right camera views. Each panel displays these synchronized camera views together with the corresponding driving instruction. The results show that LMGenDrive (1) preserves spatial consistency across views, (2) anticipates dynamic agents such as crossing vehicles and pedestrians, and (3) aligns future scene evolution with the given language instructions.

## 5 CONCLUSION

We introduced LMGenDrive, a unified framework that couples LLM-based multimodal understanding/reasoning with generative world models for closed-loop end-to-end autonomous driving. Through synergistic integration of instruction following, spatio-temporal reasoning, and realistic video generation, LMGenDrive significantly outperforms state-of-the-art methods on the CARLA LangAuto benchmark. Ablation studies verify the necessity of each core module, and results underscore the complementary benefits of unifying understanding and generation. This work offers a solid step toward embodied AGI and provides a foundation for future exploration on real-world deployment and broader cross-domain generalization.

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
