# OpenReview forum: "LMGenDrive: LLM Reasoning Meets World Models for End-to-End Driving"
_ICLR.cc/2026/Conference — ICLR 2026 Conference Withdrawn Submission_

### Official Review · Reviewer_GQeg · 2025-10-28

**Soundness:** 3
**Presentation:** 3
**Contribution:** 3
**Rating:** 6
**Confidence:** 3

**Summary:**

This paper proposes LMGenDrive, a unified framework that integrates large language model (LLM)-based reasoning with a generative world model for end-to-end autonomous driving.
The system takes multi-view driving videos and natural language instructions as inputs and produces both future visual scenes and corresponding driving actions.
The core idea is to couple the semantic reasoning ability of an LLM with the spatio-temporal consistency of a diffusion-based world model, connected through learnable world and action queries that mediate between symbolic reasoning and physical imagination.
The model is trained in three stages (vision pretraining, single-step prediction, and multi-step autoregressive refinement) and evaluated on the CARLA LangAuto benchmark, where it achieves significant improvements in Driving Score and temporal consistency over previous approaches such as LMDrive and GAIA.

**Strengths:**

- S1: Novel integration of reasoning and imagination. The paper presents one of the first concrete frameworks that unifies an LLM’s high-level reasoning with a generative world model’s physical prediction.
The proposed reasoning–imagination loop—where the LLM’s abstract plan is regularized by physically plausible video generation—addresses a long-standing gap between symbolic understanding and embodied control.
- S2: Learnable query design. The introduction of learnable world and action queries is elegant and impactful.
These queries allow the model to dynamically determine where to attend (world query) and how to act (action query), effectively bridging perception and reasoning.
Ablation results clearly show that removing these components degrades both temporal coherence and control accuracy.
- S3: Systematic multi-Stage Training Strategy. The three-stage learning pipeline is well-motivated and empirically validated.
It stabilizes training, enforces temporal consistency, and improves data efficiency compared to direct joint optimization.
- S4: Compelling empirical Gains. On CARLA’s LangAuto benchmark, LMGenDrive achieves a substantial Driving Score increase and improves both FID and FVD metrics, supporting the claim that language-guided reasoning can enhance physical world modeling.

**Weaknesses:**

- W1: Limited interpretability and analysis. The interaction between the LLM and the world model is treated as a black box. There is no visualization or probing analysis showing what the learnable queries capture or how the LLM’s reasoning influences generation.
- W2: Lack of theoretical clarity. While the reasoning–imagination loop is conceptually compelling, the paper does not formalize how LLM outputs are regularized through world-model feedback. The mechanism remains largely intuitive and lacks analytical grounding.
- W3: Evaluation scope. Experiments are restricted to CARLA and synthetic video settings.
The framework’s generalization to real-world datasets (e.g., nuScenes, Waymo) or unseen linguistic instructions is untested.
- W4: Physical consistency metrics are weak. Although FID/FVD scores improve, these metrics do not guarantee physical plausibility.
Metrics such as collision rate or rule compliance would provide stronger evidence of “physically regularized reasoning.”
- W5: Scalability concerns. The multi-stage training pipeline is computationally heavy. The paper does not discuss efficiency, convergence behavior, or deployment feasibility, which limits its practical relevance.

**Questions:**

Address W1-5.

---

### Official Review · Reviewer_Bhpo · 2025-10-29

**Soundness:** 3
**Presentation:** 3
**Contribution:** 3
**Rating:** 6
**Confidence:** 4

**Summary:**

This work focuses on the combination of an LLM-based VLA model and video generation model (world model). The LLM provides future tokens to the video generation model to generate videos, thereby enhancing the LLM's scene reasoning capabilities through video generation supervision. The training of the LLM is divided into two stages: single-step and multi-step (autoregressive). Experiments show that the proposed method achieves SOTA performance on closed-loop benchmark.

**Strengths:**

The motivation of this work is fully justified. VLA models that only predict actions or text struggle with poor understanding of 3D scenes and sparse supervision signals. Introducing video generation as an auxiliary task for model training is a reasonable design to address these two issues. Introducing a relatively decoupled conditional video generation model that receives tokens from the LLM output allows the LLM to focus primarily on scene reasoning rather than low-level image generation.

**Weaknesses:**

1. Regarding Stage 3: Multi-Step Long-Horizon Training. Do the reported ablation results compare the autoregressive multi-step trained model against a model trained with an equivalent amount of additional single-step training? A major issue in auto-regressive video generation is the accumulation of generation errors. I did not see specific designs in this paper to address this point, yet the experimental reports show significant performance gains from stage 3 training.

2. It seems that all experiments were conducted in online planning mode (feeding the model with real-collected data). The claimed capability of autoregressive offline video generation was not validated.

3. Application on real data. This work was evaluated solely within the CARLA simulator. This likely significantly reduces the difficulty of providing positive supervision via video prediction. It is recommended to supplement with experiments on real-world data, such as the NAVSIM benchmark.

**Questions:**

see weakness.

---

### Official Review · Reviewer_Wq9t · 2025-11-01

**Soundness:** 3
**Presentation:** 3
**Contribution:** 2
**Rating:** 2
**Confidence:** 3

**Summary:**

This paper tackles generalization in autonomous driving by unifying two major paradigms: LLM-based reasoning (for high-level reasoning and instruction-following capabilities) and generative world models (to imagine/generate the evolution in space and time of a scene).
The proposed model, dubbed LMGenDrive, is claimed to be the first to do this in an end-to-end model for closed-loop driving.
It uses an LLM as a central "brain" that processes language instructions and multi-view camera images (processed by a pretrained vision encoder).
The LLM processes learnable "action queries" to predict waypoints and processes "world queries" to generate a conditioning embedding for a diffusion-based video generator (the world model) . This unified design allows the model to simultaneously reason about the current scene and generate future driving videos that are consistent with the language instruction and predicted actions.
The authors propose a pogressive training recipe in 3 stages: (1) vision encoder pretraining; (2) single-step planning and generation; (3) multi-step longer horizon training  where the model predicts new video in an autoregressive manner.
LMGenDrive is evaluated in closed-loop setting on the CARLA LangAuto benchmark (language-conditioned)  achieving state-of-the-art results.

**Strengths:**

**Significance**
- This work unifies two important and promising research directions for autonomous driving: LLM-based reasoning and generative world models towards addressing long-tail generalization


**Originality**
- This unification of the LLM-based agent and the generative video world model for closed-loop driving in this format seems novel to me.
- In particular, the dual-query architecture with action queries and world queries is a novel way to have an LLM simultaneously control both planning and generation.
- The 3-stage progressive training, a form of curriculum learning towards longer-horizon fine-tuning is novel.


**Clarity**
- The paper is overall clear and easy to follow


**Quality**
- The performance gain over previous works on the LangAuto benchmark is considerable
- The ablation studies to prove the utility of all the contributions to the final proposed solution

**Weaknesses:**

**Limited validation**
- LMGenDrive is trained and evaluated only on the CARLA LangAuto benchmark, while the arguments of the authors revolve around generalization and solving real-world long-tail problems. There is no evidence of generalization to real, noisy data or even other CARLA-based benchmarks (Bench2Drive, Leaderboard 2.0). The model and conclusions may overfit to this setting. This has been already shown for LMDrive that underperformed w.r.t. SOTA methods when evaluated on other commonly used benchmarks [h]
- Some potential validations would be worth to look at Bench2Drive [a] and evaluate in the style of SimLingo [b] just the driving skills and ignore the text
- Another way to evaluate such a model also SimLingo-style to show the instruction-following capabilities/language-action alignment would be *action dreaming*, by presenting the model with the same visual scene but with different language instructions and check if the model changes accordingly
- Other ways to test such a model would be in NeuroNCAP-like [c], HUGSIM [d] or NavSim (v1 or v2) [e] evaluations on real data similarly to VaVAM [f], who also decoupled the generation head and evaluate just the driving policy. These evals can work even in single camera mode.
- For the generation part only the FID and FVD scores are reported (without any baseline) and some qualitative examples. It is currently ignored in the driving part, making it look rather an auxiliary loss than an embedded part in this unified framework that the authors described. Maybe an evaluation with a generation-only baseline with generated videos and MPC planner would make sense here.
- On LangAuto, more fine-grained scores on LangAuto can be reported as done in LMDrive (vehicle collisions, pedestrian collision, layout collision, redlight violation, etc.)
- Another setting that could be considered for training and evaluation is the CoVLA dataset [g] which has 80 hours of driving (single camera though) with trajectories and automatic captions for behavior,reasoning.


**Limited baselines**
- The method is essentially compared against other methods only on the LangAuto benchmark against 2 related methods
- However there are other related methods to consider that could better emphasize the current contributions in terms of language and world modelling. For instance SimLingo [b] which is very related and not mentioned here, or the LAW model, mentioned by the authors as the most related work.
- Alternatively the authors could implement simple variants of Vista, GAIA 1 or 2 on the currently used data and compare against that
- The generation part also has no baseline other than the ablated variants of the proposed model.


**Novelty and complexity**
- The overall pipeline, excepting the image generation part, is very similar to the one from LMDrive and I estimate that not enough credit is given to it. For instance stage 1 for pretraining the vision encoder seems identical to the one from LMDrive in terms of backbone, losses, Q-former adapter, etc.
- The contributions that stand out are the world and action queries and the video decoding, but they do stand on top of LMDrive which would deserve more credit
- The very related SimLingo [b] both in terms of approach but also on different evaluation protocols is ignored and could benefit from a discussion if a comparison is not possible
- The setting without decoding and just driving has also been previously proposed in VaVAM [f]
- I believe that the criticism of LAW for not being able to generate synthetic data is not fully fair as LMGenDrive does not show a major utilization for the generated data other than FID scores and qualitative examples. In fact, I would encourage to have a one to one comparison with LAW under similar settings.


**Performance**
- The reported scores for LMDrive are very low 10.7 DS on LangAuto vs. 36.2 reported in the original paper. Seemingly the authors used the random initialization baseline for the LLM.
- While there is an ablation of the proposed components, there are not experiments showing the sensitivity to hyper-parameter tuning, scaling in terms of data and model capacity, type of LLM architecture, etc. This limits the amount of insights that this work brings.
- Also, there are no studies on computational cost and runtime performance. The world model generator with the additional CLIP encoder, could be costly.
- There are no failure cases reported in the experiments or qualitative images.

**References:**

[a] Jia et al., Bench2drive: Towards multi-ability benchmarking of closed-loop end-to-end autonomous driving, NeurIPS 2024

[b] Renz et al., SimLingo: Vision-Only Closed-Loop Autonomous Driving with  Language-Action Alignment, CVPR 2025


[c] Ljungbergh et al., Neuroncap: Photorealistic closed-loop safety testing for autonomous driving, ECCV 2024

[d] Zhou et al., HUGSIM: A Real-Time, Photo-Realistic and Closed-Loop Simulator for Autonomous Driving, arXiv 2024

[e] Dauner et al., NAVSIM: Data-Driven Non-Reactive Autonomous Vehicle Simulation and Benchmarking, NeurIPS 2024

[f] Bartoccioni et al., VaViM and VaVAM: Autonomous Driving through Video Generative Modeling, arXiv 2025

[g] Arai et al., CoVLA: Comprehensive Vision-Language-Action Dataset for Autonomous Driving, WACV 2025

[h] Li et al., A Comprehensive Evaluation of Four End-to-End AI Autopilots Using CCTest and the Carla Leaderboard, arXiv 2025

**Questions:**

This paper takes an interesting direction of study unifying LLMs and world model video generation for end-to-end driving.  I find the endeavor of the authors nice, with a good story. However I do have several concerns regarding the limited experiments, essentially just on CARLA LangAuto, with limited comparisons against related works, missing some relevant works for the driving part and not comparing to any baseline for the video generation part. Besides, except the ablation studies, there are limited insights on the scalability of the approach in terms of data and models, the utility of different LLMs, failure modes, and the generalization to different other settings.

My current rating is leaning towards reject at this time, but I'm looking forward for the rebuttal.

Here are a few questions and suggestions that could be potentially addressed in the rebuttal or in future versions of this work (please note that suggested experiments are not necessarily expected to be conducted for the rebuttal):

1. Consider adding some baselines for the video generation part. It can be regular diffusion model (VISTA) or an autoregressive variant (e.g., GAIA-1, VaVIM) on the LangAuto dataset.

2. Extension of driving experiments to other settings to asses driving performance (Bench2Drive), vision-language alignment (action dreaming)

3. Discussion and eventual comparison with SimLingo on the driving part.

4. Extension of experiments to real data, e.g., CoVLA for pretraining, NeuroNCAP, NavSim or HUGSIM for evaluation.

5. Discussion on limitations and different failure modes.

6. Studies on the impact of the LLM model used in the framework.

---

### Note · Authors · 2025-11-13

I have read and agree with the venue's withdrawal policy on behalf of myself and my co-authors.